# 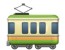 TRAMS: Training-free Memory Selection for Long-range Language Modeling

**Haofei Yu**[♡][*]**, Cunxiang Wang**[♣][†]**, Yue Zhang**[♣]**, Wei Bi**[◇][‡]
[♡]Language Technologies Institute, Carnegie Mellon University, USA
[♣]School of Engineering, Westlake University, China ◇ Tencent AI Lab, China
haofeiy@cs.cmu.edu, {wangcunxiang, zhangyue}@westlake.edu.cn,
victoriabi@tencent.com

## Abstract

The Transformer architecture is crucial for numerous AI models, but it still faces challenges in long-range language modeling. Though several specific transformer architectures have been designed to tackle issues of long-range dependencies, existing methods like Transformer-XL are plagued by a high percentage of ineffective memories. In this study, we present a plug-and-play strategy, known as **TRA**ining-free **M**emory **S**election (TRAMS), that selects tokens participating in attention calculation based on one simple metric. This strategy allows us to keep tokens that are likely to have a high attention score with the current queries and ignore the other ones. We have tested our approach on the word-level benchmark (WikiText-103) and the character-level benchmark (enwik8), and the results indicate an improvement without having additional training or adding additional parameters.

## 1 Introduction

Transformer-based models (Kenton and Toutanova, 2019; Liu et al., 2019; Raffel et al., 2020; Lan et al., 2019; Brown et al., 2020) have achieved remarkable performance over the past few years. The key component of these model architectures is the attention mechanism (Vaswani et al., 2017). However, the original attention design struggles to efficiently handle long sequences, which becomes particularly problematic in scenarios such as document-level translation (Werlen et al., 2018; Kim et al., 2019) and large-scale text generation (Zhou et al., 2023), as its time and space computation costs increase quadratically with the sequence length (Tay et al., 2022). The primary factor for this elevated computational complexity can be traced back to the multiplication between queries and keys used in

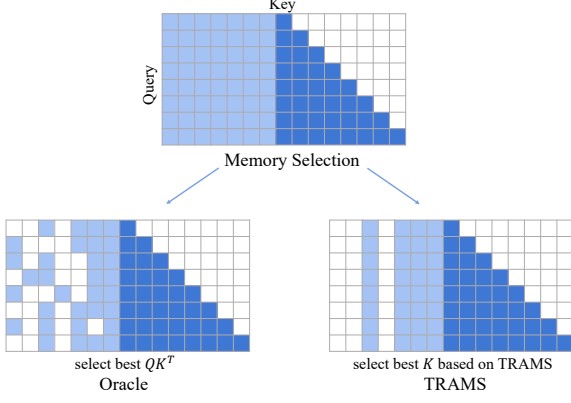

Figure 1: Two memory selection methods: For oracle, it selects memories with the highest attention scores after computing $QK^\top$. For TRAMS, it selects important key/value pairs that are independent of queries based on our self-defined metric before computing $QK^\top$.

the attention module. In general, the time complexity for calculation is $\mathcal{O}(N^2 d)$ if a transformer model with $d$ dimensions is set up with an input consisting of $N$ tokens.

To tackle this computation bottleneck, numerous efforts have been made. The first line of work is to find a new efficient expression to compute the attention score. Despite the advancements made, these methods often compromise performance, thus paving the way for alternative solutions. Efficient architectures that provide an approximate expression of attention have been explored widely (Wang et al., 2020; Peng et al., 2022b,a; Choromanski et al., 2021; Zheng et al., 2022b,a). The second line of work is to keep the calculation expression the same and use an external structure like hash function (Kitaev et al., 2019; Daras et al., 2020), clustering (Roy et al., 2021; Vyas et al., 2020) and memory selector (Pietruszka et al., 2022; Dai et al., 2019; Bertsch et al., 2023; Sukhbaatar et al., 2021, 2019; Child et al., 2019) to find the suitable subset of queries and keys in the long sequence for attention calculation.

---

[*]Work done during internship at Tencent AI Lab.
[†]Co-first Author.
[‡]The correponding author.

Our work falls into the second category, in which we propose a training-free memory selection mechanism to select suitable tokens for attention computation. Specifically, we focus on pushing Transformer-XL (Dai et al., 2019) architecture to a better position by selecting higher-quality tokens inside its memory. Based on our initial investigation, we construct a memory subset by selecting 50% of the memories with the largest attention values and maintaining the same performance. It indicates that **a large portion of information in memory is not fully utilized**. This motivates us to explore better methods to optimize memory usage.

Illustrated in Figure 1, we propose a **TRA**ining-free **M**emory **S**election method (TRAMS) that can be directly plugged into memory-based long-range language models and reduces the time complexity of computing attention matrix. Through experiments on two language modeling benchmark datasets, namely word-level `WikiText-103` (Merity et al., 2016) and character-level `enwik8` (Mahoney, 2011), we achieve an improvement in the model's performance, as demonstrated by a 0.19 perplexity (ppl) drop in `WikiText-103` and a 0.017 reduction in bits-per-character (bpc) in `enwik8`.

To our knowledge, we are the first to design a training-free memory selection method based on Transformer-XL architecture.[1]

## 2 Method

### 2.1 Problem Definition

We use $h \in \mathbb{R}^{N \times d}$ to represent the input hidden states for the attention module, $o \in \mathbb{R}^{N \times d}$ to represent the output hidden states for the attention module, $m \in \mathbb{R}^{M \times d}$ to represent the memory hidden states used in the attention calculation. We use $W_Q, W_K, W_V$ to represent the trainable projection matrix in the attention module. We define $d$ for the dimension of the model, $M$ for the memory size, and $N$ for the input size. The attention calculation process can be formally written as $o = \text{Attn}(h, m)$.

With the above annotations, the problem of memory selection can be defined as choosing a subset of hidden states memory $\tilde{m}$ from the memory $m$ that brings the minimum difference to the transformer layer output but with a smaller memory size.

$$\tilde{m}^* = \underset{\tilde{m} \subset m}{\arg\min} \|\text{Attn}(h, \tilde{m}) - \text{Attn}(h, m)\| \quad (1)$$

[1]Source code for this paper is available at https://github.com/lwaekfjlk/TRAMS.

### 2.2 Attention Reformulation

In a memory-augmented language model, the standard attention mechanism between input hidden states and memory hidden states can be written as:

$$\text{Attn}(h, m) = \text{softmax}\left(\frac{QK^\top}{\sqrt{d}}\right)V \quad (2)$$

where $Q = hW_Q$ is the product of target token hidden states $h$ and query projection matrix $W_Q$; $K = mW_K$ is the product of memory token hidden states $m$ and key projection matrix $W_K$; $V = mW_V$ is also the product of memory token hidden states $m$ and value projection matrix $W_V$.

Different from the well-known attention score calculation, we can compute this attention formula in a different order but maintain the same result:

$$\begin{aligned} QK^\top &= (hW_Q^\top)(mW_K^\top)^\top \\ &= (h)(W_Q^\top W_K m) \\ &= (h)(mW_K^\top W_Q)^\top \end{aligned} \quad (3)$$

Thus, we define $Q' = h$ as the reformulated query for this attention expression and $K' = mW_K^T W_Q$ as the reformulated keys for attention. With this reformulation, we transfer all attention-related parametric information onto reformulated key vectors.

### 2.3 Transformer Hidden Space

Since $h$ is the input of the current transformer layer and also the output of the previous transformer layer, it is the result of the last layer's `Layernorm` operation. We can define the coordinate-wise average of $h$ as $\mu$ and the coordinate-wise standard deviation of $h$ as $\sigma$. Expressions can be written as:

$$\mu = \frac{1}{d}\sum_{i=1}^{d} h_i \approx 0, \quad \sigma = \sqrt{\frac{1}{d}\sum_{i=1}^{d}(h_i - \mu)^2} \approx 1 \quad (4)$$

Since the mean value for the hidden states $h$ is around zero, we can confirm the hidden states vectors are approximately orthogonal to the $\vec{\mathbb{1}}$ vector and the L2 norm of hidden states is around $\sqrt{d}$.

With this approximation, we can expand our reformulated attention score as:

$$\begin{aligned} Q'K'^\top &= (h)(mW_K^\top W_Q)^\top \\ &= \|Q'\| \cdot \|K'\| \cdot \cos\langle Q', K'\rangle \\ &\approx \sqrt{d} \cdot \|K'\| \cdot \cos\langle Q', K'\rangle \end{aligned} \quad (5)$$

where $\|Q'\|$ stands the L2 norm for $Q'$ and $\|K'\|$ stands for the L2 norm for $K'$. Based on Fig 2,

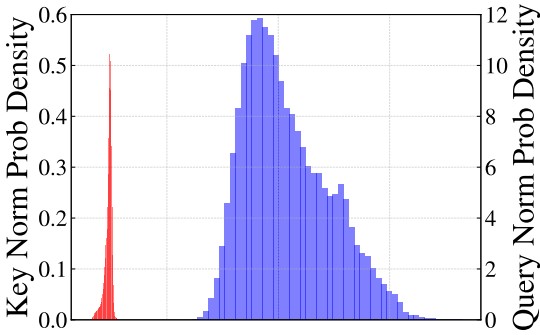

Figure 2: Norm distribution of reformulated $Q'$ and $K'$. The red distribution represents the query norm. The blue distribution represents the key norm.

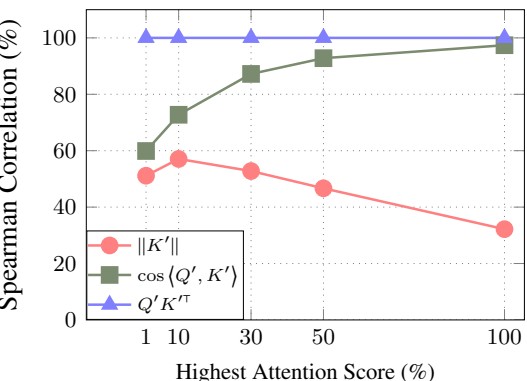

Figure 3: Spearman Correlation Score on different ranking metrics with the groundtruth one.

we see that reformulated query norm $\|Q'\|$ has a much sharper distribution compared with key norm $\|K'\|$, indicating reformulated query norm can be approximated by a constant factor.

### 2.4 Training-free Memory Selection (TRAMS)

Our target for memory selection is to recover the complete attention score with as few memory tokens as possible. This problem is equivalent to finding the subset of memory tokens that have the highest attention scores with queries. We propose a heuristic method to perform token-level selection based on a memory-independent metric in this section.

There are two crucial components for calculating the attention score after approximating $\|Q'\|$ with a constant factor: the norm of the reformulated keys $\|K'\|$ and the angles between the reformulated keys and queries $\arccos\langle Q', K'\rangle$, which is proved in Khandelwal et al. (2019). Commonly, we believe that $\arccos\langle Q', K'\rangle$ is the more important factor in general. Yet, if we use the ranking of attention score value for all query and key pairs as ground-truth ranking, based on Fig 3, we empirically discovered that rankings based on key norms and rankings based on angles produce close Spearman correlation scores when only taking the highest 1% attention scores into account. Therefore, it indicates that we can rank our memory tokens based on $\|K'\|$ solely to gain a relatively good performance when we desire top 1% attention scores with queries in our memories instead of all.

Additionally, we discovered that relying solely on a large norm isn't sufficient as a constraint. Specifically, keys that are nearer to $\vec{\mathbb{1}}$ tend to yield a higher attention score. To address this, we introduce a combined metric: $s = \cos\langle K', \vec{\mathbb{1}}\rangle \|K'\|$.

This metric allows us to identify tokens that can produce very high attention scores when paired with the appropriate query (owing to a high value of $\|K'\|$) and very low scores when paired with an unsuitable query. This is due to the near orthogonality to the query space, as indicated by a small angle with $\vec{\mathbb{1}}$, which is orthogonal to the query space.

## 3 Experiments

We introduce the compared methods and report the main results and analysis on different attention variants for inference in this section. Datasets details for `WikiText-103` and `enwik8` benchmarks and their evaluation metric details are included in Appendix A. The details of the model that we built memory selection on can be seen in Appendix B.

### 3.1 Compared Methods

**Transformer+RPE** (Vaswani et al., 2017): the vanilla transformer baseline with relative position embedding that is the same as Transformer-XL. Therefore, the only difference between this model and Transformer-XL is the additional memories. More information related to relative position embedding can be seen in Appendix C.

**Transformer-XL** (Dai et al., 2019): a specific-designed architecture for long-range language modeling. It includes relative position embedding and recurrent memories per layer. Memory slots are filled with hidden states from previous time steps.

### 3.2 Experimental Settings

We compare our methods with the Transformer-XL (Dai et al., 2019) under the same size of memory ($m = 200$) for attention calculation. For the input token length $n$ for both models, we keep the same as in (Dai et al., 2019) ($n = 64$). Additionally,

| Model | #Total Mem | #Selected Mem | PPL ($\downarrow$) |
|---|---|---|---|
| WikiText-103 | | | |
| Transformer+RPE | - | - | 29.14 |
| Transformer-XL | - | 200 | 24.17 |
| TRAMS | 600 | 200 | **23.98** |
| enwik8 | | | |
| Transformer+RPE | - | - | 1.240 |
| Transformer-XL | - | 200 | 1.215 |
| TRAMS | 600 | 200 | **1.198** |

Table 1: Model performance on the word-level `WikiText-103` and the character-level `enwik8` datasets.

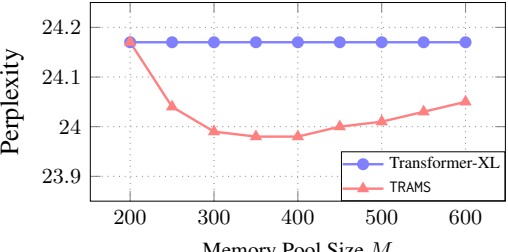

Figure 4: Ablation study on memory pool size $M$ when we fix $m$=200 and $n$=64.

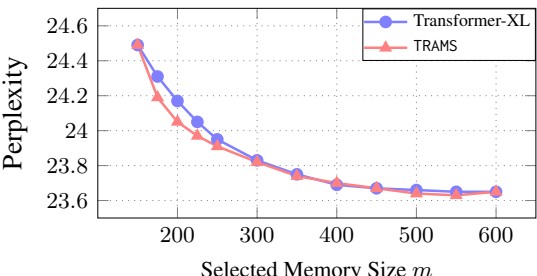

Figure 5: Ablation study on selected memory size $m$ when we fix $M$=600 and $n$=64.

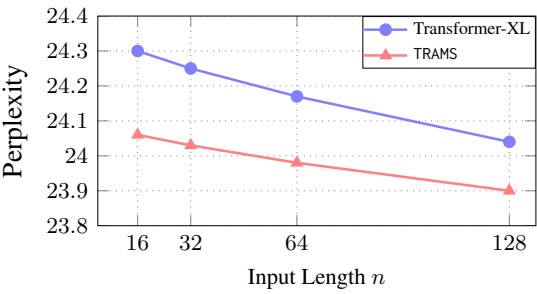

Figure 6: Ablation study on target length $n$ when we fix $M$=400 and $m$=200.

the memory selection process is performed on a memory pool with the size of $M$. Our model and the Transformer-XL share the model parameters but have different inference strategies.

### 3.3 Main Results

The main results of `WikiText-103` and `enwik8` datasets are shown in Table 1. Without additional training or additional parameters, we gain 0.19 improvement in perplexity and 0.017 improvement for bit-per-character with our TRAMS mechanism. We implement $p$-test by inferencing on multiple model checkpoints and prove that our results are significant ($p < 0.05$).

### 4 Discussions

**Is TRAMS vulnerable to the selection of hyperparameters?** There are three hyper-parameters in TRAMS: the memory pool size $M$ that TRAMS is able to select from; the selected memory size $m$ that is used in the forward process; and the input token size $n$ that is involved in both backward and forward process.

From the ablation study on $M$, Figure 4 suggests an optimal range between 300 to 400 for the memory pool size. Beyond this range, enlarging the memory pool often leads to the selection of irrelevant tokens, deteriorating our performance. Regarding $m$, Figure 5 indicates that TRAMS witnesses a substantial drop in perplexity when the memory size selected is about 25%. Selecting a larger portion does not yield further improvement. This is consistent with Figure 3, where TRAMS excels by concentrating on the top 10% of results. Lastly, in the study on $n$, Figure 6 shows that as the target token length decreases, the efficacy of memory selection improves.

**What is the inference cost compared to Transformer-XL?** Since there is no training part in our model, we focus on discussing the inference cost. Compared with Transformer-XL, our model requires storing a larger memory pool to do memory selection. Therefore, the memory cost of our method would be larger. When it comes to timing cost, our model has an additional memory token norm computation memory sorting operations, and memory selection operations for each layer. These extra operations require extra inference time. Table 2 shows the GPU memory cost and wall-clock time for the Transformer-XL baseline and our model. Our model requires slightly more GPU memory usage and around 50% additional inference time for memory selection.

| Model | Peak GPU Mem (MB) | Wall-clock Time (s) |
|---|---|---|
| Transformer-XL | 3529 | 33.27 |
| TRAMS | 3719 | 49.55 |

Table 2: Results on GPU peak memory usage and wall-clock inference time on `WikiText-103`.

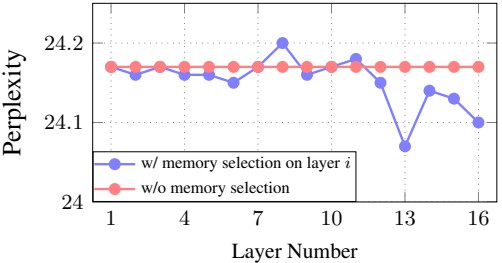

Figure 7: Ablation Study on Layer-wise Importance on `WikiText-103`.

**How does TRAMS benefit from memory selection?**
Memory selection helps the model pick tokens with higher attention scores with the queries, thus increasing the average memory utilization. Quantitatively, our method improves the average attention probability by **24.25**% for the same size of memory compared with Transformer-XL.

**Does each layer hold the same importance?**
Based on Figure 7, we show the ablation study when applying memory selection on each layer while remaining other layers the same. There is an observable drop when we apply the memory selection on the deeper layers starting from Layer 13 while we do not observe a clear influence when applying memory selection on shallow layers.

## 5 Case Study

To have an understanding of what kind of context should be selected, we provide one example case to understand specifically what kind of tokens in the memory would be selected. Based on Table 3, we can see that most of the selected memory tokens are low-frequency words. Those low-frequency words like "John" in the memory would be beneficial for the prediction of "John" in the target sequence.

## 6 Conclusion

In this work, we formulate the problem of memory selection in transformer architecture and reformulate the attention calculation process to obtain our self-defined queries and keys. After that, we propose a query-independent metric that utilizes memory hidden states to implement a training-free

**Memory Sequence Segment**

...Simon `Stephens`, which was performed in 2001 at the Royal Court Theatre. He had a guest role in the television series Judge `John` Deed in 2002. In 2004 `Boulter` landed a role as "Craig" in the episode "Teddy's Story" of the television series The Long Firm; he starred alongside actors Mark `Strong` and Derek Jacobi. He was cast in the 2005 theatre productions of the Philip Ridley play Mercury Fur, which was performed at the Drum Theatre in Plymouth and the <unk> Chocolate Factory in London. He was directed by `John` Tiffany and starred alongside Ben `Whishaw`, Shane Zaza, Harry Kent, Fraser Ayres, Sophie Stanton, and Dominic Hall. <eos> In 2006, Boulter starred alongside `Whishaw` in the play Citizenship written by Mark `Ravenhill` ...

**Target Sequence Segment**

He appeared in the television series Judge `John` Deed in 2002 ...

Table 3: Case Study for memory selection from `WikiText-103`. `text` indicates that this word in memory sequence is selected and used in the forward pass. `text` indicates that this word in the target sequence benefits from the memory.

memory selector. Our experiments indicate that this method offers a simple yet effective means of identifying valuable memory tokens. Exploring optimal memory selection strategies for large language models is a promising avenue for future research. Additionally, integrating trainable parameters into these models as memory selectors presents another exciting direction for future work.

## Limitations

Our study has a couple of main limitations. First, we are currently focusing on the Transformer-XL architecture, but there are many other models with different sizes we haven't tried. It indicates that our findings could be limited to typical transformer architecture. Second, our method has many hyperparameters including $M$, $m$, and $n$. Adjusting them can greatly change how well our model works. A careful calibration is thus necessary, and one must tread cautiously to strike a balance and achieve the desired performance, which could be time-consuming and computationally expensive.

## Ethics Statement

There are no recognized potential risks.

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

## A   Dataset and Evaluation Metrics

**WikiText-103** (Merity et al., 2016) is a commonly used word-level language modeling benchmark. It has an average length of 3.6 thousand tokens per article and includes 28 thousand Wikipedia articles. This word-level dataset has a vocabulary size of around 260K. We use the same data pre-processing setting in Dai et al. (2019) for this dataset. We use perplexity as our metric.

**Enwik8** (Mahoney, 2011) is a character-level language modeling benchmark. This dataset contains 100M unprocessed Wikipedia characters. The train set, dev set, and test set include 80M, 10M, and 10M characters separately. enwik8 has no pre-processing stage and is directly used. bpc (bit per character) is defined as an evaluation metric and we report results on both the dev set and test set.

## B   Training Configurations

Since we do inference experiments based on a trained model, we separately train two Transformer-XL models for WikiText-103 and enwik8. For the training stage, we use Adam (Kingma and Ba, 2014) to optimize with a batch size=60, learning rate=2.5e-4, target length=150, memory length=150, and a cosine scheduler without warmup steps.

When it comes to a different dataset, we use different Transformer-XL architecture. For WikiText-103, we use a 16-layer transformer architecture with 10 heads, 410 hid dim, 0.1 dropout ratio, 0.0 attention dropout ratio, 2100 inner dim, and adaptive softmax mechanism. For enwik8, we propose a 12-layer transformer architecture with 8 heads, 512 hid dim, 0.1 dropout ratio, 0.0 attention dropout ratio, and 2048 inner dim. Both models are trained for 350K steps.

A batch size=10 and target length=150 are fixed for all inference experiments to avoid unfair comparison. All experiments including training and inference are conducted using 4 2080Ti GPUs. It takes 280 GPU hours to train the enwik8 model checkpoint. It takes 61 GPU hours to train the WikiText-103 model checkpoint.

## C   Relative Position Embedding

Concerning positional encodings, we maintain the same results with Transformer-XL. The positional encodings include learnable parameters of $R_{i-j}$, $u$, and $v$. Typically, $R_{i-j}$ is derived from a learnable $r$ network included in the model. The advantage of using this design when computing the attention score is that it avoids temporal confusion caused by indexing the same position and considers the relative distance between two tokens. The formula for attention score calculation with relative position embedding can be written as:

$$A_{i,j}^{xl} = X_i^\top W_q^\top W_k^E X_j + X_i^\top W_q^\top W_k^R R_{i-j}$$
$$+ u^\top W_k^E X_j + v^\top W_k^R R_{i-j} \qquad (6)$$

Moreover, after doing ablation studies on relative position embedding, we found that $R_{i-j}$ contributes the most to the result and $u$, $v$ only has a small influence on the final performance. The existence of $R_{i-j}$ leads to the exponentially decayed attention probability distribution related to a memory position. As a result, we base our memory selection on the $A_{i,j}^{xl}$ which includes positional information instead of the pure $X_i^\top W_q^\top W_k^E X_j$. To be noticed, all concepts related to $\mathbf{qK}$ are all equipped with position embedding instead of a simple dot product.