# OpenReview forum: "TRAMS: Training-free Memory Selection for Long-range Language Modeling"
_EMNLP/2023/Conference — EMNLP 2023 Findings_

### Official Review · Reviewer_Rdm5 · 2023-08-07

**Soundness:** 4

**Excitement:**

3: Ambivalent: It has merits (e.g., it reports state-of-the-art results, the idea is nice), but there are key weaknesses (e.g., it describes incremental work), and it can significantly benefit from another round of revision. However, I won't object to accepting it if my co-reviewers champion it.

**Missing References:**

I believe that the following works are also related to this paper and would like to hear the authors opinion of how they relate to them:

A. [Generating Long Sequences with Sparse Transformers](https://arxiv.org/abs/1904.10509)

B. [Fast Transformer Decoding: One Write-Head is All You Need](https://arxiv.org/abs/1911.02150)

**Paper Topic And Main Contributions:**

The paper offers a new heuristic, based on theory and empirical that supports, for calculating which tokens should be saved in memory when using a Transformer-XL model to improve model performance without requiring any form of retraining or calibration of the model. The authors show that their new heuristic improves the model performance on 2 key datasets, WikiText-103 and enwik8.

**Questions For The Authors:**

A. How do you see this work fitting it with LLMs of today? Do you see a way to integrate any of the conclusions of your work to more modern architectures such as LLaMA, Falcon, etc?

B. Do you think that it is possible to reduce the additional GPU memory and compute time required by your heuristic? As it stands, it seems that the benefit from your method in performance is very little in comparison to the cost.

C. Why only 200 memory tokens were examined?

D. Is it possible with your method to get better results using less memory tokens?

**Reasons To Accept:**

1. Paper is well written and clear, experiments are well presented and proper statistics tests were used to strengthen the claims of the paper
2. The paper show that with proper heuristics you can improve an already pretrained model by making better use of its context memory. Also, it shows that most tokens that are saved in memory are redundant and can be discarded saving precious GPU memory and memory transactions.

**Reasons To Reject:**

1. Experiments show a very narrow image of the benefits of the proposed method. A single architecture was used with memory size of 200 tokens.
2. Unable to validate baseline results since authors used an unorthodox Tranformer-XL architecture that wasn't mentioned in the original paper
3. Insufficient comparison to alternative methods mentioned in the introduction and related work.

**Reproducibility:**

4: Could mostly reproduce the results, but there may be some variation because of sample variance or minor variations in their interpretation of the protocol or method.

**Reviewer Confidence:**

2: Willing to defend my evaluation, but it is fairly likely that I missed some details, didn't understand some central points, or can't be sure about the novelty of the work.

**Typos Grammar Style And Presentation Improvements:**

Table 1: The first row model field (Transformer+RPE (Vaswani et al., 2017)) is out of bounds and make the table unclear for reading.

Code link is not working

---

> ### Author Rebuttal · Authors · 2023-08-29
>
> We thank the reviewers for providing valuable suggestions and encouraging us to provide more details related to the experiments.
>
> # Rebuttal Point1
>
> **_To respond to the first reason to reject:_**
>
> **_“Experiments show a very narrow image of the benefits of the proposed method. A single architecture was used with memory size of 200 tokens.”_**
>
> Thank you for highlighting this concern. We'd like to address the mentioned issues in the following aspects:
>
> **A Single Architecture**: We believe Transformer-XL is a well-known architecture in the long-context language modeling field, comparing all related works. Thus, choosing Transformer-XL as the baseline can better show the efficiency of our method. Given the page limit of a short paper, we want to show the most important results in our current experiment section. TRAMS could potentially be layered atop more advanced versions of Memory Transformers, such as LaMemo and Infinity-Former. If our work is accepted with one more page, we are willing to add such results.
>
> **Memory Size Sensitivity**: As for the memory size, we have added some sensitivity analysis. Please kindly refer to our answer (Part 2) to question B for Reviewer ncPF (**Rebuttal Point4 for Reviewer ncPF**) to check the comprehensive analysis.
>
> **Case Study**: To give a broader image of our benefits beyond the quantitative experimental details, we have also undertaken qualitative experiments in the form of case studies. Specifically, we wish to shed light on the types of memory tokens that are predominantly chosen during the memory selection process. **Two examples randomly sampled from the wikitext-103 dataset are shown at the end of our rebuttal.** Based on our case studies, we observed that named entities such as "John Deed" and "Du Fu" are more likely to be selected over high-frequency words like "You," "he," and "me." This observation underlines how our method enhances performance. Specifically, low-frequency entities within the target tokens benefit from memory tokens that are also low-frequency entities. This suggests that our memory selection mechanism not only improves model performance but also adds an interpretative layer to the model's decisions.This observation underlines how our method enhances performance. Specifically, low-frequency entities within the target tokens benefit from memory tokens that are also low-frequency entities. This suggests that our memory selection mechanism not only improves model performance but also adds an interpretative layer to the model's decisions.
>
> [1] Haozhe Ji et. al: LaMemo: Language Modeling with Look-Ahead Memory [[https://arxiv.org/abs/2204.07341](https://arxiv.org/abs/2204.07341)]
>
> [2] Zihang Dai et al: Transformer-XL: Attentive Language Models Beyond a Fixed-Length Context [[https://arxiv.org/abs/1901.02860](https://arxiv.org/abs/1901.02860)]
>
> [3] Pedro Henrique Martins et al: ∞-former: Infinite Memory Transformer [[https://arxiv.org/abs/2109.00301](https://arxiv.org/abs/2109.00301)]
>
> # Rebuttal Point2
>
> **_To respond to the second reason to reject (part1):_**
>
> **_“(Unable to validate baseline results since) authors used an unorthodox Tranformer-XL architecture that wasn't mentioned in the original paper”_**
>
> Thank you for bringing up the concern regarding the validity of our baseline reproduction.
>
> **Comparison of Architecture:** Firstly, it's important to note that the reformulated transformer attention proposed in our paper maintains the same parameters and architectural elements as the standard Transformer-XL model. The distinction lies solely in the computation order.
>
> **Computation Order Change:** In the standard Transformer-XL, hidden states for each layer are initially mapped as Key and Query. Following this, the model calculates the dot product between the Key and Query. However, in our reformulated attention mechanism, we project the hidden states based on  $W_k$  and  $W_q$ , allowing us to calculate  $hW_k^TW_q$  before proceeding to compute the dot product with the original hidden states $h$.
>
> **Purpose of the Conceptual Change:** This reordering in computation enables us to develop our unique scoring method based solely on $hW_k^TW_q$, rather than requiring both $hW_q^T$  and $hW_k^T$. In essence, this adjustment is a conceptual modification rather than an architectural one. Through this conceptual change, we can naturally introduce a scoring function that relies on the norm computation of a single vector $hW_k^TW_q$.
>
> # Rebuttal Point3
>
> **_To respond to the second reason to reject (part2):_**
>
> **_“Unable to validate baseline results (since authors used an unorthodox Tranformer-XL architecture that wasn't mentioned in the original paper)”_**
>
> Thanks for pointing out this important baseline validation problem.
>
> To respond to the baseline validation, this question is accidentally overlapped with reject reason one mentioned by reviewer ncPF (**Rebuttal Point1 for Reviewer ncPF**). We provide thorough experimental results (Table 1 and Table 2 mentioned in response to the first reason to reject in review1) for fair comparison. This serves to validate the performance of our baseline model.
>
> # Rebuttal Point4
>
> **_To respond to the third reason to reject:_**
>
> **_“Insufficient comparison to alternative methods mentioned in the introduction and related work.”_**
>
> Thank you for highlighting the need for a more exhaustive comparison with alternative methods in the introduction and related works. We organize alternative methods we found besides those mentioned in the paper during the rebuttal period. We would have a more comprehensive comparison for the introduction and related works in our final version.
>
> **Method Classification:** Recent developments in this field have been abundant, prompting us to categorize related works into distinct approaches. The first general strategy aims to create more efficient Transformer architectures capable of accommodating longer context inputs. The second employs additional components, such as token or memory corpuses, to enhance the model's ability to capture long-distance dependencies. We present some alternative methods discovered during the rebuttal period for further context.
>
> (Tay et al.) provides a clear roadmap for different Transformer architecture variants. In its roadmap, Transformer-XL is classified as one type of recurrence-augmented Transformer architecture.
>
> **Extending Context Window:** Tay et al. offer a comprehensive roadmap delineating various Transformer architecture variants. Within this framework, Transformer-XL is categorized as a recurrence-augmented type. Numerous works like Funnel Transformer (Dai et al.), Poolingformer (Zhang et al.), Longformer (Beltagy et al.), Big Bird (Zaheer et al.), and ETC (Ainslie et al.) propose methods centered around memory selection or downsampling to create more adaptive Transformer architectures. These can be fine-tuned for tasks requiring longer context windows. Additionally, innovative projects like Random Feature Attention (Peng et al.), Linear Randomized Attention (Zheng et al.), and EVA (Zheng et al.) focus on optimizing attention modules within the Transformer framework. Recent advancements such as Focused Transformer (Tworkowski et al.) and Position Interpolation (Chen et al.) leverage well pre-trained large language models to extend context windows. These initiatives enable the models to assimilate more information from extended contextual windows.
>
> **Utilizing External Module:** Another avenue to enhance long-distance dependency recognition involves external memory modules. KNN-LM (Khandelwal et al.) and Memorizing Transformer (Wu et al.) utilize k-nearest-neighbor algorithms to bolster the capabilities of the original Transformer model. Unlimiformer (Bertsch et al.) employs a modified version of the Memorizing Transformer to extend the model's context capacity indefinitely with the aid of external memory storage.
>
> **State Space Research:** A related but distinct research trajectory that deserves mention focuses on the use of structured state spaces explicitly designed for long-context modeling (Gu et al.).
>
>
>
> **Alternative Methods:**
>
> [1] Yi Tay et al. Efficient Transformer: A Survey [[https://arxiv.org/abs/2009.06732](https://arxiv.org/abs/2009.06732)]
>
> [2] Zihang Dai et al. Funnel-Transformer: Filtering out Sequential Redundancy for Efficient Language Processing [[https://arxiv.org/pdf/2006.03236.pdf](https://arxiv.org/pdf/2006.03236.pdf)]
>
> [3] Hang Zhang et al. Poolingformer: Long Document Modeling with Pooling Attention [[https://arxiv.org/pdf/2105.04371.pdf](https://arxiv.org/pdf/2105.04371.pdf)]
>
> [4] Iz Beltagy et al. Longformer: The Long-Document Transformer [[https://arxiv.org/pdf/2004.05150.pdf](https://arxiv.org/pdf/2004.05150.pdf)]
>
> [5] Manzil Zaheer et al. Big Bird: Transformers for Longer Sequences [[https://arxiv.org/abs/2007.14062](https://arxiv.org/abs/2007.14062)]
>
> [6] Joshua Ainslie et al. ETC: Encoding Long and Structured Inputs in Transformers [[https://arxiv.org/pdf/2004.08483.pdf](https://arxiv.org/pdf/2004.08483.pdf)]
>
> [7] Szymon Tworkowski et al. Focused Transformer: Contrastive Training for Context Scaling [[https://arxiv.org/abs/2307.03170](https://arxiv.org/abs/2307.03170)]
>
> [8] Shouyuan Chen et al. Extending Context Window of Large Language Models via Positional Interpolation [[https://arxiv.org/abs/2306.15595](https://arxiv.org/abs/2306.15595)]
>
> [9] Hao Peng et al. Random Feature Attention [[https://arxiv.org/abs/2103.02143](https://arxiv.org/abs/2103.02143)]
>
> [10] Lin Zheng et al. Efficient Attention via Control Variates [[https://arxiv.org/abs/2302.04542](https://arxiv.org/abs/2302.04542)]
>
> [11] Lin Zheng et al. Linear Complexity Randomized Self-attention Mechanism [[https://arxiv.org/abs/2204.04667](https://arxiv.org/abs/2204.04667)]
>
> [12] Albert Gu et al. Efficiently Modeling Long Sequences with Structured State Spaces [[https://arxiv.org/pdf/2111.00396.pdf](https://arxiv.org/pdf/2111.00396.pdf)]
>
> [13] Urvashi Khandelwal et al. Generalization through Memorization: Nearest Neighbor Language Models [[https://arxiv.org/abs/1911.00172](https://arxiv.org/abs/1911.00172)]
>
> [14] Amanda Bertsch et al. Unlimiformer: Long-Range Transformers with Unlimited Length Input [[https://arxiv.org/abs/2305.01625](https://arxiv.org/abs/2305.01625)]
>
> [15] Yuhuai Wu et al. Memorizing Transformers [[https://arxiv.org/abs/2203.08913](https://arxiv.org/abs/2203.08913)]
>
> # Rebuttal Point5
>
> **_To answer Question A:_**
>
> **_A. How do you see this work fitting it with LLMs of today? Do you see a way to integrate any of the conclusions of your work to more modern architectures such as LLaMA, Falcon, etc?_**
>
> Thank you for your insightful comments on the integration of TRAMS and LLMs.
>
> Our model can still be useful in the era of LLMs for the following reasons:
>
> **Easy Implementation with No Extra Training Required:** A common challenge in adapting new methods to LLMs is that they often necessitate additional training designs. Our approach, however, eliminates this need. Specifically designed to focus on memory selection, our TRAMS method can be directly plugged into current LLMs for either pre-training, supervised finetuning, or only the inference stage.
>
> **Memory Selection Remains Crucial:** Current LLMs need to allow long context as inputs and long text as outputs. Both features need to involve potentially large attention matrix computations. Therefore, it is important to investigate memory selection techniques such as our work,  to allow useful information to be selected from the long context/generated output.
>
> **Improved Representational Capabilities of LLMs**: The increased potency of LLMs suggests that they offer a superior representational space compared to smaller models like the Transformer-XL. With a more structured semantic landscape, we have a better chance of performing selections based on individual representation vectors without having to rely on query representation.
>
> It can be a promising direction for our future work.
>
> # Rebuttal Point6
>
> **_To answer Question B:_**
>
> **_“Question B. Do you think that it is possible to reduce the additional GPU memory and compute time required by your heuristic? As it stands, it seems that the benefit from your method in performance is very little in comparison to the cost.”_**
>
> Thanks for pointing out the issue of additional cost.
>
> **Space and Time Complexity:** To explain Table 2 in the original paper in more detail, the main difference between TRAMS and Transformer-XL is that , for space complexity, TRAMS requires a larger memory pool compared with Transformer-XL for each layer before the attention mechanism. For time complexity, TRAMS requires additional computation of the norm of each reformulated key vector when compared with Transformer-XL. Therefore, TRAMS brings an extra space complexity of  $\mathcal{O}((M-m) \times L)$, where  $M$ represents overall memory pool size for TRAMS,  $m$ represents memory length for Transformer-XL and $L$ stands for the number of layers. TRAMS brings an extra time complexity of $\mathcal{O}(M \times L)$, where $M$ represents memory pool size and $L$ stands for the number of layers.
>
> However, we think it is possible to reduce the additional GPU memory and compute time required by my heuristic. Therefore, we propose two methods that can reduce the GPU memory and compute time:
>
> **Selective Layer Memory Selection:** As indicated in Figure 4 of our paper, the last few layers exert a greater influence on performance than the preceding layers. One strategy to improve is to perform memory selection only on the last few layers. This change reduces the time complexity from $\mathcal{O}(m \times L)$ to $\mathcal{O}(m \times l)$, where $l$ denotes the selected layers and $L$ represents all the transformer layers.
>
>
>
> Table 10: GPU and Wall-Clock Time for Selective Layer Memory Management
>
> | Model Setting                                 | GPU Peak Memory (MB) | Wall-clock Time (s) | Perplexity |
> | --------------------------------------------- | -------------------- | ------------------- | ---------- |
> | Transformer-XL(mentioned in the paper)        | 3529                 | 33.27               | 24.17      |
> | TRAMS (on last five layers)                   | 3590                 | 43.95               | 24.00      |
> | TRMS (on all layers) (mentioned in the paper) | 3719                 | 49.55               | 23.98      |
>
> **Dynamic Online Memory Selection:** Another method to further minimize GPU memory usage involves dynamic memory management. We can select the top-$k$ tokens from the target length tokens and maintain a dynamic list of these top-$k$ tokens. It reduces the time complexity from $\mathcal{O}(m)$ to $\mathcal{O}(n)$, where $m$ stands for memory length and $n$ represents the target length. We leave the implementation and discussion of this online memory selection method as part of our future work due to limited rebuttal time.
>
> We will include this discussion in the final version.
>
> # Rebuttal Point7
>
> **_To answer Question C:_**
>
> **_Question C. Why only 200 memory tokens were examined?_**
>
>  Please kindly refer to our answer to question B in reviewer ncPF (**Rebuttal Point4 for Reviewer ncPF**).
>
> # Rebuttal Point8
>
> **_To answer Question D:_**
>
> **_Question D. Is it possible with your method to get better results using less memory tokens?_**
>
> Thanks for pointing out a valuable direction for future work.
>
> **Conclusion:** It is not easy to accomplish it with the current TRAMS method. First, the Transformer-XL architecture is sensitive to a small memory (like around 100). However, a better scoring function for memory selection may help, which is promising for further exploration. We provide a detailed analysis below:
>
>
>
> Table11: Discussion when continuously decreasing the memory size
>
> | Target Token Length | Memory Pool Size | Selected Memory Size/Actual Used Memory Size | Transformer-XL PPL(↓) | TRAMS PPL(↓) | TRAMS PPL drops compared with Transformer-XL (↑) |
> | ------------------- | ---------------- | -------------------------------------------- | --------------------- | ------------ | ------------------------------------------------ |
> | 64                  | 600              | 200                                          | 24.17                 | 24.05        | 0.12                                             |
> | 64                  | 600              | 160                                          | 24.41                 | 24.27        | 0.24                                             |
> | 64                  | 600              | 100                                          | 25.08                 | 24.95        | 0.13                                             |
>
> **Transformer-XL Sensitivity**: From Table 11, we find that when the memory size changes from 200 to 160, the TRAMS perplexity change increases by 0.22 points. The TRAMS method can use memory selection to improve performance, but we cannot select too few tokens. The reason is that when the memory size is very small, like 100, the nearest tokens are almost the most useful tokens for generation. The benefits of extracting one token from the context history are smaller than those of remaining with the nearest token.
>
> **Future Work**: We believe that there should be a way to develop a better scoring function for memory selection such that we can select fewer memory tokens and get better performance at the same time. For example, we can directly optimize the set selection problem, which is to identify a subset of memory tokens that maximizes the remaining attention value. However, it is not trivial to find a solution for this optimization problem and we leave it as our future work.
>
> # Rebuttal Point9
>
> **_To respond to missing reference:_**
>
> **_“Missing References:_**
>
> **_I believe that the following works are also related to this paper and would like to hear the authors opinion of how they relate to them:_**
>
> **_A. Generating Long Sequences with Sparse Transformers_**
>
> **_B. Fast Transformer Decoding: One Write-Head is All You Need”_**
>
> Thanks for providing the missing references, we will include them in the revised paper. Furthermore, we will provide a more comprehensive comparison with the two papers in the revised paper.
>
> When it comes to the relationship between our paper and theirs, both papers propose new Transformer architectures for longer and faster generation tasks. Both of them are super interesting and related to our paper’s topic.
>
> The “**Sparse Transformer**” paper also focuses on memory selection via masking and targets at providing a more efficient transformer architecture. One major difference is that Sparse Transformer is trying to give different masks for each different query. Moreover, the selected memory directly participates in the forward-backward training process. Therefore, it is a sparse architecture instead of a “memory selection” process that selects side information in an additional part of the original model.
>
> The “**Fast Transformer Decoding**” paper proposes a variant of attention called multi-query attention that shares a single set of keys and values. Similar to ours, this paper tries to propose a new attention architecture. The major difference between ours and theirs is that our motivation is the memory redundancy phenomenon discovered in Transformer-XL architecture. Their motivation is that limited memory-bandwidth would cause repeated loading of key/value tensors and slow inference.
>
>
>
> # Rebuttal Appendix
>
> We listed two case study examples that are mentioned in the **Rebuttal for current Reviewer** here.
>
> **Case1:**
>
> **Sequence segments that in the memory pool (top-k memory tokens are emphasized here):**
>
> _Simon **Stephens** , which was performed in 2001 at the Royal Court Theatre . He had a guest role in the television series Judge **John** Deed in 2002 . In 2004 **Boulter** landed a role as " Craig " in the episode " Teddy 's Story " of the television series The Long Firm ; he starred alongside actors Mark **Strong** and Derek Jacobi . He was cast in the 2005 theatre productions of the Philip Ridley play Mercury Fur , which was performed at the Drum Theatre in Plymouth and the &lt;unk> Chocolate Factory in London . He was directed by **John** Tiffany and starred alongside Ben **Whishaw** , Shane Zaza , Harry Kent , Fraser Ayres , Sophie Stanton and Dominic Hall . &lt;eos> In 2006 , Boulter starred alongside **Whishaw** in the play Citizenship written by Mark **Ravenhill**. He appeared on a 2006 episode of the television series , Doctors , followed by a role in the 2007 theatre production of How to Curse directed by Josie Rourke . How to Curse was performed at Bush Theatre in the London Borough of Hammersmith and Fulham . Boulter starred in two films in 2008 , Daylight Robbery by filmmaker **Paris** **&lt;unk>**, and Donkey Punch directed by Olly Blackburn . In May 2008 , Boulter made a guest appearance on a two @-@ part episode arc of the television series Waking the **Dead** , followed by an appearance on the television series Survivors in November 2008 . He had a recurring role in ten episodes of the television series Casualty in 2010 , as " Kieron Fletcher " . Boulter starred in the 2011 film Mercenaries directed by Paris &lt;unk> . &lt;eos> &lt;eos> = = Career = = &lt;eos> &lt;eos> &lt;eos> = = = 2000 – 2005 = = = &lt;eos> &lt;eos> In 2000 Boulter had a guest @-@ starring role on the television series The Bill ; he portrayed " Scott Parry " in the episode , " In Safe Hands " . Boulter starred as " Scott " in the play Herons written by Simon Stephens , which was performed in 2001 at the Royal Court Theatre . A review of Boulter 's performance in The Independent on Sunday described him as " horribly menacing " in the role , and he received critical reviews in The Herald , and Evening Standard . He appeared_
>
> **Sequence segments that we are handling (target tokens that can be potentially benefited from the selected memories are highlighted):**
>
> _in the television series Judge **John Deed** in 2002 as " **&lt;unk>** Armitage " in the episode " Political &lt;unk> " , and had a role as a different character " Toby Steele " on The Bill . &lt;eos> He had a recurring role in 2003 on two episodes of The Bill , as character " Connor Price " . In 2004 **Boulter** landed_
>
> **Case2:**
>
> **Sequence segments that in the memory pool (top-k memory tokens are emphasized here):**
>
> _criticized old commentaries **during** _the _Yuan Dynasty were too unfathomable . Matsuo **Bashō** , the greatest haiku poet , was also strongly influenced by Du Fu ;**in** Oku no **Hosomichi** , his masterpiece , he cites the first two lines of A Spring **View** ( &lt;unk> **)** before a haiku as its introduction and also many of his other haiku have similar wording and themes . It is said that when he died in Osaka **during** a long travel , a copy of Du Fu 's poetry was found with him as one of a few precious items which he was able to carry around . **&lt;eos> &lt;eos>** = = Translation = = &lt;eos> &lt;eos> A variety of styles have been used in efforts to translate **Du Fu** 's work into **English** . As Burton Watson remarks in The Selected Poems of Du **Fu** , " There are many different ways to approach the problems involved in translating Du Fu , which is why we need as many different translations as possible " ( p. &lt;unk> ) . The translators have **had** to contend with bringing out the formal constraints of the original without sounding laboured to a Western ear ( particularly when translating regulated **verse** , or &lt;unk> ) , and accommodating the complex allusions contained particularly in the later works ( Hawkes writes that " his poems do not as a rule come through very well in translation " — p. ix ) . One extreme on each issue is represented by Kenneth Rexroth 's One Hundred Poems From the Chinese . His are free translations , which seek to conceal the &lt;unk> through &lt;unk> and expansion and contraction of the content ; his responses to the allusions are firstly to omit most of these poems from his selection , and secondly to " translate out " the references in those works which he does select . &lt;eos> Other translators have placed much greater weight on trying to convey a sense of the poetic forms used by Du Fu . Vikram Seth in Three Chinese Poets uses English @-@ style rhyme schemes , whereas Keith Holyoak in Facing the Moon approximates the Chinese rhyme scheme ; both use end @-@ stopped lines and preserve some degree of parallelism . In The Selected Poems of Du Fu , Burton Watson follows the &lt;unk> quite strictly , persuading the_
>
> **Sequence segments that we are handling (target tokens that can be potentially benefited from the selected memories are highlighted):**
>
> _western reader to adapt to the poems rather than vice versa . Similarly , he deals with the allusion of the later works by combining literal translation with extensive annotation . &lt;eos> In 2015 , Stephen Owen published translations , with facing Chinese texts , of the complete poetry of **Du Fu** in six volumes , with extensive scholarly apparatus , which emphasized &lt;unk>_

---

### Official Review · Reviewer_XG1k · 2023-08-08

**Paper Topic And Main Contributions:** 1. Introduction of a novel approach o…
**Soundness:** 3

**Excitement:**

3: Ambivalent: It has merits (e.g., it reports state-of-the-art results, the idea is nice), but there are key weaknesses (e.g., it describes incremental work), and it can significantly benefit from another round of revision. However, I won't object to accepting it if my co-reviewers champion it.

**Questions For The Authors:**

See Reasons To Reject.  I will reconsider my score in the rebuttal.

**Reasons To Accept:**

1. **Comprehensive Experimental Setup**: The paper provides a detailed experimental setup which allows readers to understand the conditions under which the proposed method was tested. This ensures reproducibility and gives credibility to the results presented.
2. **Comparison with State-of-the-art**: The authors have taken the effort to compare their method with existing techniques. This is crucial for the research community to understand the relative advantages of the new method.
3. **Clarity**: From the sections provided, it appears that the authors have made an effort to be clear in their presentation, making it accessible to readers who might be new to the topic.

**Reasons To Reject:**

1. **Incomplete Content**: Given that only a portion of the paper is available, it's challenging to understand the full context, especially the theoretical foundations and the broader implications of the results. This makes it hard to gauge the complete impact of the paper.
2. **Lack of Detailed Discussion**: While the experimental setup is comprehensive, there seems to be a lack of in-depth discussion on the results. Such a discussion would provide insights into why certain observations were made and the implications of the results.
3. **Potential Over-reliance on Experimental Results**: While empirical evaluations are important, it would be beneficial if the paper also provided theoretical justifications or proofs for the proposed method. This would strengthen the paper's claims and make the results more robust.

**Reproducibility:**

3: Could reproduce the results with some difficulty. The settings of parameters are underspecified or subjectively determined; the training/evaluation data are not widely available.

**Reviewer Confidence:**

3: Pretty sure, but there's a chance I missed something. Although I have a good feel for this area in general, I did not carefully check the paper's details, e.g., the math, experimental design, or novelty.

---

> ### Author Rebuttal · Authors · 2023-08-29
>
> # Rebuttal Point1
>
> **_To respond to the first reason to reject:_**
>
> **_“Incomplete Content: Given that only a portion of the paper is available, it's challenging to understand the full context, especially the theoretical foundations and the broader implications of the results. This makes it hard to gauge the complete impact of the paper.”_**
>
> Regarding the **full context** of our paper, we are primarily driven by the observed memory redundancy in the Transformer-XL model. Our main contribution is the introduction of a novel memory selection method tailored for models that utilize transformer attention mechanisms and external working memory during the inference phase. These motivations and contributions are clearly articulated in the introduction, detailed in the methodology section, and their effectiveness is substantiated in the experimental section.
>
> On the topic of **theoretical foundations**, we have framed the Transformer memory selection challenge as an optimal set selection optimization task. Our proposed TRAMS method is designed as an approximation algorithm to address this optimization challenge. While we acknowledge that our approximation might not come with a theoretical performance guarantee, its efficacy is evident from our empirical results. We'd like to emphasize that our submission is to a conference that focuses on "empirical" methods in NLP. As such, we believe it would be disproportionate to dismiss our work solely based on the absence of theoretical guarantees.
>
> Lastly, concerning the **broader implications** of our findings, our research introduces a training-free, plug-and-play technique. This is significant, especially when considering that previous studies have seldom treated memory selection as a standalone function rooted in key/value vectors. Our methodology provides a fresh perspective on the representation spaces molded by attention mechanisms. We are confident that our research will pave the way for further innovative studies in this area.
>
> # Rebuttal Point2
>
> **_To respond to the second reason to reject:_**
>
> **_“Lack of Detailed Discussion: While the experimental setup is comprehensive, there seems to be a lack of in-depth discussion on the results. Such a discussion would provide insights into why certain observations were made and the implications of the results.”_**
>
> Firstly, we would like to clarify the depth of our current discussions. Our paper encompasses a thorough analysis, including topics such as _"how the model benefits from memory selection", "the comparative inference cost with Transformer-XL", and an exploration into whether "each layer holds equal importance"_. We are somewhat perplexed about the specific areas of discussion deemed lacking. We kindly request more explicit feedback on the specific discussions or analyses that the reviewer found absent. We are more than willing to undertake additional experiments or discussions to address these gaps. For instance, Reviewer ncPF expressed concerns about the sensitivity of our configuration setting. To address this, we have incorporated relevant results. We direct the reviewer to our response to question B in Reviewer ncPF's section, where we delve into a detailed ablation study concerning hyper-parameter selection.
>
> # Rebuttal Point3
>
> **_To respond to the third reason to reject:_**
>
> **_“Potential Over-reliance on Experimental Results: While empirical evaluations are important, it would be beneficial if the paper also provided theoretical justifications or proofs for the proposed method. This would strengthen the paper's claims and make the results more robust.”_**
>
> Frankly speaking, our approach is rooted in intuitive reasoning, focusing on entity-based tokens at the semantic space's extremities. Theoretically establishing a gap boundary is challenging for our context. We've conducted rigorous empirical evaluations, aligning with the conference's focus on empirical methods in NLP. While theoretical proofs are valuable, our empirical validations offer significant insights in the dynamic domain of NLP. We believe our empirical approach provides robust contributions to the field.

---

### Official Review · Reviewer_ncPF · 2023-08-09

**Soundness:** 4

**Excitement:**

3: Ambivalent: It has merits (e.g., it reports state-of-the-art results, the idea is nice), but there are key weaknesses (e.g., it describes incremental work), and it can significantly benefit from another round of revision. However, I won't object to accepting it if my co-reviewers champion it.

**Paper Topic And Main Contributions:**

The authors proposed a training-free memory selector which targets improving the memory efficiency of Transformer-XL (Dai et al., 2019)'s. The contribution of this work includes introducing a ranking-based method, selecting top-k memory tokens via $\text{cos} \langle K', \vec{\mathbb{1}}\rangle ||K'||$, where $||K'||$ is the reformulated key derived in equation 4. According to the results in Table 1, applying the proposed memory selector to the Transformer-XL reduces the PPL of the Standard Transformer-XL by 0.19 on the WikiText-103 dataset and the BPC of a 12L Transformer-XL by 0.017 on the enwik8 dataset respectively, with the memory size set to 200.

**Questions For The Authors:**

- $\textbf{Question A}$: As mentioned in "Reasons to reject", what is the PPL of TRAMS when the memory size=150? Same question to enwiki8, what is the BPC of TRAMS when the memory size=512?
- $\textbf{Question B}$: Following the above question, is TRAMS sensitive to "Mem Size"? Are there any reasons for setting "Mem Size" to 200?
- $\textbf{Question C}$:  In L68-L70, "only 25% of the top memories are essential for maintaining the performance". As 25% comes from WikiText-103, does the proportion of redundant memory dataset specific?
- $\textbf{Question D}$(optional): In Section 2.4, how the number of selected memory tokens is decided after ranking? Is it a fixed number? Just curious why `topk_num` is set to 50 in L356 (https://anonymous.4open.science/r/TRAMS-768A/utils/modeling_transfo_xl.py#L356).

**Reasons To Accept:**

1. The paper provides constructive information which gives readers a clear view of how the proposed method TRAMS is formed.
2. The proposed memory-selection algorithm is simple to plug into existing transformers-based models, which is novel and effective in improving memory efficiency.

**Reasons To Reject:**

1. Table 1 needs more experimental results to convince readers that the proposed method TRAMS improves Transformer-XL. For example, in the WikiText-103 dataset, the PPL of 12L Transformer-XL 24.17 differs from the reported PPL 24.0 in the Transformer-XL (Dai et al., 2019) paper. Please refer to "Ours - Transformer-XL Standard" in Table 1 of Dai's work. The dissimilarity comes from different memory sizes, 200 in this work and 150 in the original Transformer-XL (see run_wt103_base.sh: https://github.com/kimiyoung/transformer-xl/blob/master/pytorch/run_wt103_base.sh). Experiments on the proposed method TRAMS with the same memory size as Transformer-XL by datasets are required.

**Reproducibility:**

4: Could mostly reproduce the results, but there may be some variation because of sample variance or minor variations in their interpretation of the protocol or method.

**Reviewer Confidence:**

4: Quite sure. I tried to check the important points carefully. It's unlikely, though conceivable, that I missed something that should affect my ratings.

---

> ### Author Rebuttal · Authors · 2023-08-29
>
> We thank the reviewers for providing valuable suggestions and encouraging us to provide more details related to the experiments.
>
> # Rebuttal Point1
>
> **_To respond to the first reason to reject:_**
>
> **_“Table 1 needs more experimental results to convince readers that the proposed method TRAMS improves Transformer-XL. For example, in the WikiText-103 dataset, the PPL of 12L Transformer-XL 24.17 differs from the reported PPL 24.0 in the Transformer-XL (Dai et al., 2019) paper. Please refer to "Ours - Transformer-XL Standard" in Table 1 of Dai's work. The dissimilarity comes from different memory sizes, 200 in this work and 150 in the original Transformer-XL (see run_wt103_base.sh: https://github.com/kimiyoung/transformer-xl/blob/master/pytorch/run_wt103_base.sh). Experiments on the proposed method TRAMS with the same memory size as Transformer-XL by datasets are required.”_**
>
> Thank you for bringing up the important issue of fair baseline comparison in our research. When preparing for the final version of our paper, we would put the target token length and memory token length number as part of our experimental setting section to avoid confusing people on that. We would add this comparison in the main experimental table to show that our paper is working on a successfully reproduced baseline Transformer-XL checkpoint.
>
> **Token Length Roles:** The discrepancy in Perplexity (PPL) values in the Wikitext-103 dataset can be attributed primarily to variations in two settings: target token length and memory token length. The target token length relates to the number of tokens that participate actively in the backward process of the model. Memory token length, on the other hand, means tokens that are omitted from this backward process and have stop-gradient operations applied. These differences form the basis for varying PPL values.
>
> **Reproduced Baselines:** To validate the implementation of our Transformer-XL baseline model, originally developed by Dai et al., we undertook a two-pronged approach. Firstly, we meticulously aligned our model's configurations with the settings outlined in the original Transformer-XL paper. Secondly, we evaluated our model's performance against the official Transformer-XL baseline reproduced in LaMemo.
>
> **Reproduced Baselines Details:** The original Transformer-XL model was trained with 150 target tokens and 150 memory tokens. During inference, however, the model deploys 64 target tokens and 640 memory tokens—a shift from the 150+150 memory + target token lengths used during training. This setting is documented in the [Transformer-XL repository](https://github.com/kimiyoung/transformer-xl/blob/master/pytorch/run_wt103_base.sh). Conversely, the LaMemo model by Ji et al. maintains a consistent setting for both training and inference, employing 150 target tokens and 150 memory tokens as mentioned in the section 4.3 in their original paper. Given these divergent configurations, we felt it necessary to evaluate our model against both these established baselines.
>
> **Experimental Results:** We display the results of our experiments with the replicated Transformer-XL model in Table1 and Table2. Our findings indicate that our re-implemented model performs at par with, or even exceeds, the baselines established in both the Dai et al. and Ji et al. papers. These results are consistent across the enwik8 and Wikitext-103 datasets. As a result, we can assert with confidence that the improvements observed in our version of the model are indeed valid.
>
> Table1: Performance comparison between different token length model settings on Wikitext-103 dataset
>
> | Model Setting                    | Target Token Length | Memory Token Length | PPL(↓)    |
> | -------------------------------- | ------------------- | ------------------- | --------- |
> | Transformer-XL (Dai et al. 2019) | 64                  | 640                 | 24.0      |
> | Transformer-XL (our reproduced)  | 64                  | 640                 | **23.66** |
> | Transformer-XL (Ji et al. 2022)  | 150                 | 150                 | 24.56     |
> | Transformer-XL (our baseline)    | 150                 | 150                 | **24.26** |
> | Transformer-XL (our baseline)    | 64                  | 200                 | **24.17** |
>
> Table2: Performance comparison between different token length model settings on enwik-8 dataset
>
> | Model Setting                    | Target Token Length | Memory Token Length | BPC(↓)    |
> | -------------------------------- | ------------------- | ------------------- | --------- |
> | Transformer-XL (Dai et al. 2019) | 80                  | 2100                | **1.06**  |
> | Transformer-XL (our reproduced)  | 80                  | 2100                | 1.094     |
> | Transformer-XL (Ji et al. 2022)  | 512                 | 512                 | 1.239     |
> | Transformer-XL (our reproduced)  | 512                 | 512                 | **1.122** |
> | Transformer-XL (our reproduced)  | 64                  | 200                 | 1.215     |
>
> [1] Zihang Dai et. al. Transformer-XL: Attentive Language Models Beyond a Fixed-Length Context [[https://arxiv.org/abs/1901.02860](https://arxiv.org/abs/1901.02860)]
>
> [2] Haozhe Ji et. al. LaMemo: Language Modeling with Look-Ahead Memory [[https://arxiv.org/abs/2204.07341](https://arxiv.org/abs/2204.07341)]
>
> # Rebuttal Point2
>
> **_To answer question A:_**
>
> **_“Question A: As mentioned in "Reasons to reject", what is the PPL of TRAMS when the memory size=150? Same question to enwiki8, what is the BPC of TRAMS when the memory size=512?”_**
>
> That is a very good question. I think this question is crucial.
>
> **Memory Pool Size Influence:** The performance of TRAMS is not solely dependent on a fixed memory size but also hinges on the pool size from which memory tokens are selected. For instance, with a memory size of 150, TRAMS utilizes 150 memory tokens for inference. However, the source of these tokens—whether they are drawn from a pool of 300 or more—can affect performance. Given these fluctuations, we find it essential to discuss TRAMS' performance in relation to different sizes of memory pools. Therefore, below we show two tables discussing the performance of our method in different memory pool sizes.
>
> **Wikitext-103 Results:** Table 3 varies the size of the memory pool, which has a significant impact on perplexity. A memory pool size of 350 appears to be the most effective. When the memory pool size is too small, perplexity doesn't decrease substantially. This is intuitive, as a smaller pool offers fewer candidate tokens for selection. On the other hand, when the memory pool size is too large, perplexity also fails to decrease appreciably. This is likely because a larger pool may include too many irrelevant tokens that are far removed from the current context, leading the model to make incorrect selections more often.
>
> **Enwik8 Results:** Table 4 compares different ratios of selected memory size to actual used memory size. We observe that TRAMS consistently outperforms the Transformer-XL baseline when the ratio is set at 512, regardless of the target token size. However, as the memory pool size increases—reaching as large as 1536—we notice a decline in performance compared to the Transformer-XL baseline. Interestingly, a "U-shaped" curve emerges when we analyze the effects of memory pool size on the enwik8 dataset as well. This indicates that on enwik8 dataset, it is important to find a suitable memory pool size for memory selection. One noticeable thing is that the performance improvement of TRMAS with 512 memory size is not as good as the number mentioned in our paper. We think the reason is because enwik8, as a character-level LM task, has a limited vocab, and it is difficult for TRAMS to find typically useful tokens when having 512 available memories.
>
>
>
> Table3: Performance comparison with different memory pool sizes for memory selection in TRAMS for wikitext-103 dataset
>
> | Settings       | Memory Pool Size | Selected Memory Size/Actual Used Memory Size | Target Token Size | PPL(↓)    |
> | -------------- | ---------------- | -------------------------------------------- | ----------------- | --------- |
> | Transformer-XL | —                | 150                                          | 150               | 24.26     |
> | TRAMS          | 200              | 150                                          | 150               | 24.14     |
> |                | 250              | 150                                          | 150               | 24.08     |
> |                | 300              | 150                                          | 150               | 24.07     |
> |                | 350              | 150                                          | 150               | **24.06** |
> |                | 400              | 150                                          | 150               | 24.09     |
> |                | 450              | 150                                          | 150               | 24.11     |
> |                | 500              | 150                                          | 150               | 24.13     |
> |                | 550              | 150                                          | 150               | 24.16     |
> |                | 600              | 150                                          | 150               | 24.19     |
>
> Table4: Performance comparison with different memory pool sizes for memory selection in TRAMS for enwik8 dataset
>
> | Settings       | Memory Pool Size | Selected Memory Size/Actual Used Memory Size | Target Token Size | BPC(↓)    |
> | -------------- | ---------------- | -------------------------------------------- | ----------------- | --------- |
> | Transformer-XL | —                | 512                                          | 512               | 1.122     |
> | TRAMS          | 1024             | 512                                          | 512               | **1.117** |
> |                | 1536             | 512                                          | 512               | 1.123     |
> | Transformer-XL | —                | 512                                          | 80                | 1.137     |
> | TRAMS          | 1024             | 512                                          | 80                | **1.132** |
> |                | 1536             | 512                                          | 80                | 1.144     |
>
> # Rebuttal Point3
>
> **_To answer question B (part1):_**
>
> **_“Question B: Following the above question, is TRAMS sensitive to "Mem Size"?”_**
>
> The simple answer is **not** sensitive to “memory size”. Below is a detailed illustration.
>
> **“Mem Size” Definition:** The “Meme size” mentioned in our paper means “selected memory size / actual used memory size” in the TRAMS. This term refers to the subset of tokens selected from a larger pool of nearest memory tokens for use during the inference process. In contrast, in the Transformer-XL model, "selected memory size/ actual used memory size" pertains to the nearest memory tokens themselves.
>
> **Sensitivity Test:** In response to the previous question, we explored the sensitivity of performance to changes in memory pool size. Now, we shift our focus to understanding how varying the selected memory size impacts performance, while keeping the memory pool size constant. From the findings presented in Table 5, as referred to in answer to question B, both TRAMS and Transformer-XL exhibit similar trends as the memory size changes. Specifically, when the memory size is on the smaller side—ranging from 150 to 200—both models experience noticeable fluctuations in performance. In contrast, when the memory size is larger—within the 300 to 400 range—performance changes are relatively negligible.
>
> **TRAMS vs Transformer-XL:** Upon examining the performance curves for TRAMS and Transformer-XL in Table 5, it becomes clear that TRAMS experiences a steeper decline in perplexity (PPL) as memory size increases. For instance, as the memory size rises from 150 to 200, PPL falls by 0.32 for Transformer-XL, as opposed to a more significant drop of 0.52 for TRAMS. Conversely, when the memory size escalates from 350 to 400, Transformer-XL sees only a modest 0.06 decline in PPL, while TRAMS remains stable, showing no change.
>
> We think sensitivity is definitely an important part of our method. We would add one subsection at the discussion section to answer the question "Is our method sensitive to memory size?".
>
> # Rebuttal Point4
>
> **_To answer question B (part2):_**
>
> **_“Are there any reasons for setting "Mem Size" to 200?”_**
>
> Thank you for pointing out the need for clarification of setting "Mem Size" to 200.
>
> **Reason for 200:** The choice of using 200 memory tokens as a hyper-parameter is primarily heuristic. We noticed a significant performance drop when pairing a target token length of 64 with a memory token length of 200. As a result, we decided to retain these hyper-parameters for our experiments.
>
> To provide further insights into this heuristic selection, we conducted an ablation study with additional experimental results:
>
> 1. **Impact of Memory Pool Size (Table 5):** Initially, we experimented with varying the memory pool sizes while keeping the selected memory length constant.
> 2. **Impact of Selected Memory Size (Table6):** Secondly, we varied the selected memory size while maintaining a fixed selection ratio.
> 3. **Impact of Absolute Memory Size (Table7):** Next, we experimented with the same selection ratio but altered the absolute memory length to assess its impact.
> 4. **Importance of Target Token Lengths (Table 8):** Finally, we examined the importance of different target token lengths by varying them while keeping the memory pool size and selected memory size constant.
>
>
>
> Table5: Ablation Study when having fixed memory pool size and different selected memory size
>
> | Target Token Length | Memory Pool Size | Selected Memory Size/Actual Used Memory Size | Selection Ratio | Transformer-XL PPL(↓) | TRAMS PPL(↓) | TRAMS PPL Drops compared with Transformer-XL (↑) |
> | ------------------- | ---------------- | -------------------------------------------- | --------------- | --------------------- | ------------ | ------------------------------------------------ |
> | 64                  | 600              | 150                                          | 25.0%           | 24.49                 | 24.49        | 0                                                |
> | 64                  | 600              | 175                                          | 29.17%          | 24.31                 | 24.19        | **0.12**                                         |
> | 64                  | 600              | 200                                          | 33.3%           | 24.17                 | 24.05        | **0.12**                                         |
> | 64                  | 600              | 225                                          | 37.5%           | 24.05                 | 23.97        | 0.08                                             |
> | 64                  | 600              | 250                                          | 41.7%           | 23.95                 | 23.91        | 0.04                                             |
> | 64                  | 600              | 300                                          | 50.0%           | 23.83                 | 23.82        | 0.01                                             |
> | 64                  | 600              | 350                                          | 58.3%           | 23.75                 | 23.74        | 0.01                                             |
> | 64                  | 600              | 400                                          | 66.7%           | 23.69                 | 23.70        | -0.01                                            |
> | 64                  | 600              | 450                                          | 75%             | 23.67                 | 23.67        | 0                                                |
> | 64                  | 600              | 500                                          | 83.3%           | 23.66                 | 23.64        | 0.02                                             |
> | 64                  | 600              | 550                                          | 91.7%           | **23.65**             | **23.63**    | 0.02                                             |
> | 64                  | 600              | 600                                          | 100.0%          | **23.65**             | 23.65        | 0                                                |
>
> Table6: Ablation Study when having fixed selected memory size and different memory pool size
>
> | Target Token Length | Memory Pool Size | Selected Memory Size/Actual Used Memory Size | Selection Ratio | Transformer-XL PPL(↓) | TRAMS PPL(↓) | TRAMS PPL Drops compared with Transformer-XL (↑) |
> | ------------------- | ---------------- | -------------------------------------------- | --------------- | --------------------- | ------------ | ------------------------------------------------ |
> | 64                  | 200              | 200                                          | 100%            | 24.17                 | 24.17        | 0                                                |
> | 64                  | 250              | 200                                          | 80.0%           | 24.17                 | 24.04        | 0.13                                             |
> | 64                  | 300              | 200                                          | 66.7%           | 24.17                 | 23.99        | 0.18                                             |
> | 64                  | 350              | 200                                          | 57.14%          | 24.17                 | **23.98**    | **0.19**                                         |
> | 64                  | 400              | 200                                          | 50%             | 24.17                 | **23.98**    | **0.19**                                         |
> | 64                  | 450              | 200                                          | 44.4%           | 24.17                 | 24.00        | 0.17                                             |
> | 64                  | 500              | 200                                          | 40%             | 24.17                 | 24.01        | 0.16                                             |
> | 64                  | 550              | 200                                          | 36.4%           | 24.17                 | 24.03        | 0.14                                             |
> | 64                  | 600              | 200                                          | 33.3%           | 24.17                 | 24.05        | 0.12                                             |
>
> Table7: Ablation Study when having fixed selection ratio but different memory pool sizes and selected memory size
>
> | Target Token Length | Memory Pool Size | Selected Memory Size/Actual Used Memory Size | Selection Ratio | Transformer-XL PPL(↓) | TRAMS PPL(↓) | TRAMS PPL Drops compared with Transformer-XL (↑) |
> | ------------------- | ---------------- | -------------------------------------------- | --------------- | --------------------- | ------------ | ------------------------------------------------ |
> | 64                  | 400              | 200                                          | 50%             | 24.17                 | 23.98        | **0.19**                                         |
> | 64                  | 800              | 400                                          | 50%             | 23.69                 | **23.83**    | -0.14                                            |
> | 64                  | 1200             | 600                                          | 50%             | **23.65**             | 23.99        | -0.34                                            |
> | 64                  | 1600             | 800                                          | 50%             | 23.72                 | 24.24        | -0.52                                            |
>
>
>
> Table8: Ablation Study when having different target lengths but with fixed memory pool size and selected memory size
>
> | Target Token Length | Memory Pool Size | Selected Memory Size/Actual Used Memory Size | Selection Ratio | Transformer-XL PPL(↓) | TRAMS PPL(↓) | TRAMS PPL Drops compared with Transformer-XL (↑) |
> | ------------------- | ---------------- | -------------------------------------------- | --------------- | --------------------- | ------------ | ------------------------------------------------ |
> | 16                  | 400              | 200                                          | 50%             | 24.30                 | 24.06        | **0.24**                                         |
> | 32                  | 400              | 200                                          | 50%             | 24.25                 | 24.03        | 0.22                                             |
> | 64                  | 400              | 200                                          | 50%             | 24.17                 | 23.98        | 0.19                                             |
> | 128                 | 400              | 200                                          | 50%             | **24.04**             | **23.90**    | 0.14                                             |
>
>
>
> **Table 5 Insights:**
>
> The benefits of selecting more tokens to the current TRAMS model diminish as more tokens are included. The top-k selected tokens are most impactful during the inference process when k is relatively small. This finding aligns with the results in Figure 3 of the paper.
>
> **Table 6 Insights:**
>
> When using a larger memory pool, there seems to be an optimal level of selection ratio, around 50%, that yields the best performance. To be noticed, the selection ratio is selecting a shared key/value vector for all queries. The selection ratio mentioned by Figure 1 in the paper is selecting unique key/value vector sets for each unique query. With too few memory tokens, it does not provide many choices. With too many, the tokens become outdated and less relevant to the current context. This results in a "U-shaped" curve in terms of perplexity reduction.
>
> **Table 7 Insights:**
>
> When the selection ratio is held constant, only the top-k tokens (when k is small) contribute significantly to the model's performance. This suggests that the attention distribution in Transformer-XL is rather sharp than flat.
>
> **Table 8 Insights:**
>
> As the target token length decreases, the token selection process becomes more effective. This can be attributed to the fact that longer target token lengths amplify the influence of position embedding, which can negatively impact the token selection process and make the model make mistakes when choosing the correct tokens.
>
> These ablation studies offer nuanced insights into various aspects of token selection and memory pool dynamics in TRAMS and Transformer-XL models. We will include these discussions in the revised paper upon acceptance. When having an extra page for the final version, we would draw graphs for those tables to allow reader easy to understand the sweet spot phenomenon and tendency when trying to select hyper-parameters.
>
> # Rebuttal Point5
>
> **_To answer question C:_**
>
> **_“Question C: In L68-L70, "only 25% of the top memories are essential for maintaining the performance". As 25% comes from WikiText-103, does the proportion of redundant memory dataset specific?”_**
>
> We'd like to clarify this issue and I think your suggestion is very useful.
>
> **Clarification on 25% Top Memory Selection Ratio:** Firstly, we have to clarify an ambiguous point in our paper regarding the selection of top memories for each query token. Assume we have 3 query tokens \{ $Q_1$, $Q_2$, $Q_3$ \} and 4 key/value tokens \{ $KV_1$, $KV_2$, $KV_3$, $KV_4$ \}. We pick the best key/value tokens for each query token—such as  $KV_2$  and  $KV_3$  for  $Q_1$ , $KV_3$ and  $KV_4$ for $Q_2$, and $KV_1$  and $KV_2$  for $Q_3$. We conclude that only 25% of top memories are actually useful. This conclusion is straightforward as it merely requires the computation of the attention map [query_len $\times$ key\_len] matrix and the selection of top-$n$ elements in each row.
>
> **TRAMS Memory Selection:** Our proposed TRAMS method simplifies this process. For the same 3 query tokens and 4 key/value tokens, TRAMS selects \( $KV_2$, $KV_3$ \), and all queries \( $Q_1$, $Q_2$, $Q_3$ \) attend to these two key-value vectors. Instead of computing a full attention map, we compute the norms of all key candidates using our reformulated attention mechanism. While we may not achieve identical performance when using only 25% of the memory, we observe improved performance compared to the Transformer-XL baseline with equivalent memory usage. This shows that our memory selection mechanism positively impacts model performance.
>
> **Memory Selection Comparison:** There are two ways to do memory selection.
>
> The first one is selecting tokens with highest attention scores (need to compute attention scores). The second one is TRAMS (not need to compute attention scores)
>
> - **Selecting tokens with highest attention scores:**
>
>   - Compute $ QK^T $ matrix
>
>   - Choose a unique key set index for each query token
>
> - **TRAMS:**
>   - Compute the norm of $K'$  using our reformulated attention mechanism
>
>   - Choose one key set index for all query tokens
>
> **Experimental Setup:** We conducted our experiments on the enwik8 dataset to evaluate the performance of computing attention scores in contrast to the TRAMS.
>
>
>
> Table9: Ablation Study on memory redundancy in enwik8 dataset
>
> | target length | overall available memory length | attention score-based memory length | Top Attention Ratio | TRAMS BPC(↓) |
> | ------------- | ------------------------------- | ----------------------------------- | ------------------- | ------------ |
> | 80            | 600                             | 50                                  | 8.30%               | 1.144        |
> | 80            | 600                             | 100                                 | 16.67%              | 1.135        |
> | 80            | 600                             | 200                                 | 33.33%              | 1.128        |
> | 80            | 600                             | 300                                 | 50.00%              | 1.125        |
> | 80            | 600                             | 400                                 | 66.67%              | 1.124        |
> | 80            | 600                             | 600                                 | 100%                | 1.123        |
>
> **Memory Redundancy Ratio Comparison:** Based on the data in Table 9, the optimal proportion of redundant memory size varies depending on the dataset. For the enwik8 dataset, for instance, the ideal level of redundancy is from 16% to 50%. This variation can be attributed to differing levels of **token diversity** across datasets. Take the wikitext-103 dataset, which is used for word-level language modeling tasks, for example. This dataset has a vocabulary size of 267,735, encompassing all the words that appear within it. On the other hand, the enwik8 dataset is used for character-level language modeling and has a much smaller vocabulary size of just 205, including all characters in the dataset. Therefore, the disparity in token diversity between the datasets affects the efficacy of our method, leading to different optimal memory selection ratios for each.
>
> We will include these discussions in the revised paper upon acceptance. When having an extra page for the final version, we would add a graph of enwik8 dataset memory redundancy in the introduction section to create a stronger motivation.
>
> # Rebuttal Point6
>
> **_To answer question D:_**
>
> **_“Question D(optional): In Section 2.4, how the number of selected memory tokens is decided after ranking? Is it a fixed number? Just curious why topk_num is set to 50 in L356 ([https://anonymous.4open.science/r/TRAMS-768A/utils/modeling_transfo_xl.py#L356](https://anonymous.4open.science/r/TRAMS-768A/utils/modeling_transfo_xl.py#L356)).”_**
>
> Thank you for raising the question about implementation details.
>
>
>
> **Ablation Study for Memory Selection:** The number of memory tokens we chose for our experiments was guided by empirical testing. Our underlying intuition suggests that both extremely small and large memory selection ratios are unlikely to provide significant gains over the original Transformer-XL baseline. For a systematic analysis detailing the impact of various hyper-parameters, readers can refer to Tables 5, 6, 7, and 8.
>
>
>
> **Code Implementation:** When it comes to the code implementation, the approach to memory use during inference is bifurcated into two segments. To illustrate, let's assume a Transformer-XL model with a target token length of 150 and an available memory pool size of 600. From this pool, we might opt to select 200 tokens. This two-step process is outlined as follows:
>
> 1. Initial Selection: Firstly, we directly select the nearest 150 tokens from the memory pool, aligning with the target length of 150.
> 2. Subsequent Selection: Then, we pick an additional 50 tokens from the remaining 450-memory-token pool. This is why the code includes a parameter topk_num=50.
>
>
>
> **Influence of Positional Embeddings:** The rationale for this particular method of memory token selection stems from the Transformer-XL's architecture, specifically its use of relative positional embeddings. To visualize, let's represent the relative position embedding for the nth token as POS_n. The pattern for these embeddings would look like:
>
> | M3   | M2   | M1   | T1   | T2   | T3   | T4   |
> | ---- | ---- | ---- | ---- | ---- | ---- | ---- |
> | POS4 | POS3 | POS2 | POS1 | x    | x    | x    |
> | POS5 | POS4 | POS3 | POS2 | POS1 | x    | x    |
> | POS6 | POS5 | POS4 | POS3 | POS2 | POS1 | x    |
> | POS7 | POS6 | POS5 | POS4 | POS3 | POS2 | POS1 |
>
> In this arrangement, the position embeddings for the first few "target length" tokens in the memory (like M1, M2, and M3) carry significant positional information. Neglecting to include these tokens would be imprudent given their importance in the model's calculations.
>
> We will include these discussions in the revised paper upon acceptance. When having an extra page for the final version, we would add a positional embedding influence heatmap graph at the appendix. Moreover, we would add a subsection in the discussion section and discuss how many nearest token should we choose.

---

### Meta-Review · Area_Chair_1twW · 2023-09-18

**Recommendation:** 3

**Metareview:**

This paper proposes a training-free memory selector to improve memory efficiency in Transformer-XL models, which includes a ranking-based method to select the top-k memory tokens. The proposed method, TRAMS, is demonstrated to be effective in improving memory efficiency through experiments.

Strengths:
* The paper is well-structured and clearly explains the proposed method.
* TRAMS is a novel and effective memory-selection algorithm that is easy to integrate with existing transformer-based models.
* The experimental setup is comprehensive and the authors compare their method with state-of-the-art techniques.

Weaknesses:
* There is a lack of detailed discussion on the results and their implications.
* The paper would benefit from more experimental results and theoretical justifications to support the claims.
* Incomplete content makes it difficult to understand the full context and impact of the paper.

Overall speaking, this paper gives some insightful notes on the method and has clear goodness. However, details and many contents need to be enhanced.

---

### Decision · Program_Chairs · 2023-10-07

**Decision:**

Accept-Findings

**Comment:**

This paper proposes a training-free memory selector to improve memory efficiency in Transformer-XL models, which includes a ranking-based method to select the top-k memory tokens. The proposed method, TRAMS, is demonstrated to be effective in improving memory efficiency through experiments.

Strengths:
* The paper is well-structured and clearly explains the proposed method.
* TRAMS is a novel and effective memory-selection algorithm that is easy to integrate with existing transformer-based models.
* The experimental setup is comprehensive and the authors compare their method with state-of-the-art techniques.

Weaknesses:
* There is a lack of detailed discussion on the results and their implications.
* The paper would benefit from more experimental results and theoretical justifications to support the claims.
* Incomplete content makes it difficult to understand the full context and impact of the paper.

Overall speaking, this paper gives some insightful notes on the method and has clear goodness. However, details and many contents need to be enhanced.